# Enhancing Transformation-Based Defenses Against Adversarial Attacks with a Distribution Classifier

**Connie Kou**[1,2]**, Hwee Kuan Lee**[1,2,3,4]**, Ee-Chien Chang**[1]**, Teck Khim Ng**[1]
[1]School of Computing, National University of Singapore
[2]Bioinformatics Institute, A*STAR Singapore
[3]Image and Pervasive Access Lab (IPAL), CNRS UMI 2955
[4]Singapore Eye Research Institute
{koukl,changec,ngtk}@comp.nus.edu.sg, leehk@bii.a-star.edu.sg

## Abstract

Adversarial attacks on convolutional neural networks (CNN) have gained significant attention and there have been active research efforts on defense mechanisms. Stochastic input transformation methods have been proposed, where the idea is to recover the image from adversarial attack by random transformation, and to take the majority vote as consensus among the random samples. However, the transformation improves the accuracy on adversarial images at the expense of the accuracy on clean images. While it is intuitive that the accuracy on clean images would deteriorate, the exact mechanism in which how this occurs is unclear. In this paper, we study the distribution of softmax induced by stochastic transformations. We observe that with random transformations on the clean images, although the mass of the softmax distribution could shift to the wrong class, the resulting distribution of softmax could be used to correct the prediction. Furthermore, on the adversarial counterparts, with the image transformation, the resulting shapes of the distribution of softmax are similar to the distributions from the clean images. With these observations, we propose a method to improve existing transformation-based defenses. We train a separate lightweight distribution classifier to recognize distinct features in the distributions of softmax outputs of transformed images. Our empirical studies show that our distribution classifier, by training on distributions obtained from clean images only, outperforms majority voting for both clean and adversarial images. Our method is generic and can be integrated with existing transformation-based defenses.

## 1 Introduction

There has been widespread use of convolutional neural networks (CNN) in many critical real-life applications such as facial recognition (Parkhi et al., 2015) and self-driving cars (Jung et al., 2016). However, it has been found that CNNs could misclassify the input image when the image has been corrupted by an imperceptible change (Szegedy et al., 2013). In other words, CNNs are not robust to small, carefully-crafted image perturbations. Such images are called adversarial examples and there have been active research efforts in designing attacks that show the susceptibility of CNNs. Correspondingly, many defense methods that aim to increase robustness to attacks have been proposed.

Stochastic transformation-based defenses have shown considerable success in recovering from adversarial attacks. Under these defenses, the input image is transformed in a certain way before feeding into the CNN, such that the transformed adversarial image would no longer be adversarial. As the transformation is random, by feeding in samples of the transformed image through the CNN, we accumulate a set of CNN softmax outputs and predictions. As such, existing transformation-based defenses take a majority vote of the CNN predictions from the randomly transformed image (Prakash et al., 2018; Guo et al., 2017). Transformation-based defenses are desirable as there is no need to retrain the CNN model. However, they suffer from deterioration of performance on clean images. With increasing number of pixel deflections (Prakash et al., 2018), there is improvement on

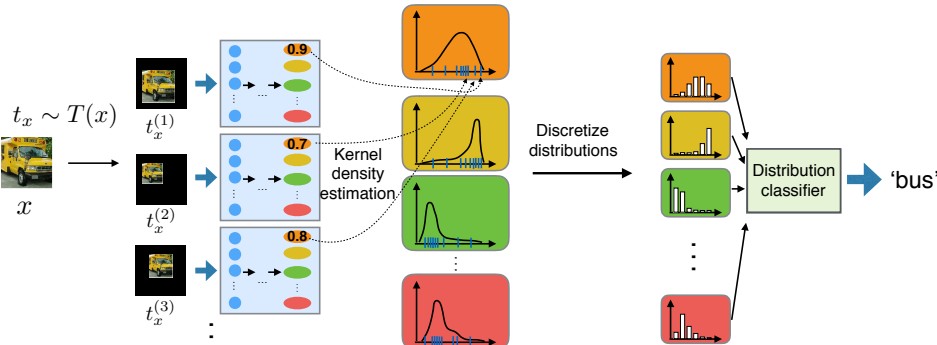

Figure 1: In transformation-based defenses, the image is transformed stochastically where each sample $t_x$ is drawn from the distribution $T(x)$ and then fed to the CNN (blue box). In our defense method, for each input image $x$, we build the marginal distribution of softmax probabilities from the transformed samples $t_x^{(1)}, t_x^{(2)}, \cdots$. The distributions are fed to a separate distribution classifier which performs the final classification. Note that our distribution classifier is trained only on distributions obtained from clean images while tested on both clean and adversarial images.

the performance on adversarial images, but this comes with a rapid deterioration of performance on clean images.

The exact mechanism of the deterioration in performance on clean images is unclear. We believe that the softmax distribution induced by the random transformation contains rich information which is not captured by majority vote that simply counts the final class predictions from the transformed samples. Now, an interesting question is whether the features in the distribution of softmax could be better utilized. In this paper, to elucidate how the deterioration in accuracy on clean images occurs, we study the effects of the random image transformations on the distribution of the softmax outputs and make some key observations. After the image transform, some clean images show distributions of softmax with modes at an incorrect class, reflecting the deterioration in voting accuracy as observed before. While the shifting of the distribution mode to the incorrect class is detrimental to the voting prediction, the resulting distribution of softmax contains features that is useful for correcting the prediction. In addition, we observe that the adversarial counterparts show similar shifts in the distributions of softmax as the clean images. We also look into the distribution shapes for the transformed clean and adversarial images and find that they are similar.

With these observations, we propose a simple method to improve existing transformation-based defenses, as illustrated in Figure 1. We train a separate lightweight distribution classifier to recognize distinct features in the distributions of softmax outputs of transformed clean images and predict the class label. Without retraining the original CNN, our distribution classifier improves the performance of transformation-based defenses on both clean and adversarial images. On the MNIST dataset, the improvements in accuracy over majority voting are 1.7% and 5.9% on the clean and adversarial images respectively. On CIFAR10, the improvements are 6.4% and 3.6% respectively. Note that the distributions obtained from the adversarial images are not included in the training of the distribution classifier. In real-world settings, the type of attack is not known beforehand. Training the distribution classifier on a specific attack may cause the classifier to overfit to that attack. Hence, it is an advantage that our defense method is attack-agnostic. Our experimental findings show that the features of the distribution in the softmax are useful and can be used to improve existing transformation-based defenses. Our contributions are as follows:

1. We analyze the effects of image transformation in existing defenses on the softmax outputs for clean and adversarial images, with a key finding that the distributions of softmax obtained from clean and adversarial images share similar features.

2. We propose a method that trains a distribution classifier on the distributions of the softmax outputs of transformed clean images only, but show improvements in both clean and adversarial images. This method is agnostic to the attack method, does not require retraining of the CNN and can be integrated with existing transformation-based methods.

## 2 Related work: Attacks and defenses

Given an image dataset $\{(x_1, y_1) \cdots (x_M, y_M)\}$ and a classifier $F_\theta$ that has been trained with this dataset with parameters $\theta$, the aim of the attack is to produce an adversarial image $x_i^{adv}$ such that $F_\theta(x_i^{adv}) \neq y_i$, and $||x_i^{adv} - x_i||$ is small. We focus on four gradient-based untargeted attacks. The Fast Gradient Sign Method (FGSM) (Goodfellow et al., 2014) is a single-step attack that uses the sign of the gradient of the classification loss to perturb the image. Iterative Gradient Sign Method (IGSM) (Kurakin et al., 2016a) is an iterative version of FGSM. In DeepFool (Moosavi-Dezfooli et al., 2016), at each iteration, the attack approximates the classifier with a linear decision boundary and generates the minimal perturbation to cross the boundary. Finally, the Carlini & Wagner (C&W) (Carlini & Wagner, 2017) $L_2$ attack jointly minimizes the perturbation $L_2$ norm and a differentiable loss function based on the classifier's logit outputs. Besides gradient-based attacks, there are also black-box attacks where the CNN model is not known and only the softmax output or final prediction is given (Brendel et al., 2017; Ilyas et al., 2018; Cheng et al., 2018).

Defense methods have been proposed to make the classifiers more robust. In adversarial training, the CNN model is trained on adversarial examples generated from itself (Madry et al., 2017; Kurakin et al., 2016b) or from an ensemble of models (Tramèr et al., 2017). Other methods involve training auxiliary neural networks on mixture of clean and adversarial images, for instance, by denoising the inputs with a neural network before feeding into the CNN (Liao et al., 2018; Song et al., 2017; Samangouei et al., 2018) or by training a neural network on the CNN logits (Li et al., 2019). In the next section, we introduce another class of defense: transformation-based defenses.

### 2.1 Transformation-based defenses

Transformation-based defenses aim to recover from adversarial perturbations, that is for input transformation $T$, we want $F_\theta(T(x_i^{adv})) = y_i$. At the same time, the accuracy on the clean images has to be maintained, ie. $F_\theta(T(x_i)) = y_i$. Note that transformation-based defenses are implemented at test time and this is different from training-time data augmentation. Here we introduce two transformation-based defenses that we experiment on.

**Pixel deflection (PD) (Prakash et al., 2018) :** Pixel deflection corrupts an image by locally redistributing pixels. At each step, it selects a random pixel and replaces it with another randomly selected pixel in a local neighborhood. The probability of a pixel being selected is inversely proportional to the class activation map (Zhou et al., 2016). Lastly, there is a denoising step based on wavelet transform. In our experiments, we did not use robust activation maps for our datasets as we found that this omission did not cause significant difference in performance (see Appendix D.3).

**Random resize and padding (RRP) (Xie et al., 2017) :** Each image is first resized to a random size and then padded with zeroes to a fixed size in a random manner.

In many transformation-based methods, the transformation is stochastic. Hence there can be different samples of the transformation of an image: $t_x \sim T(x)$, where $t_x$ represents a transformed sample. Existing transformation defenses benefit from improved performance by taking the majority vote across samples of random transformations. The advantage of transformation-based methods is that there is no retraining of the CNN classifier. However, a weakness, as identified by Prakash et al. (2018), is that the transformation increases the accuracy on adversarial images at the expense of the accuracy on clean images. The exact mechanism of the deterioration in performance on clean images is unclear. In this paper, we elucidate how the deterioration in accuracy on clean images occurs by studying the effects of the random image transformations on the distribution of the softmax outputs.

## 3 Analysis on distributions of softmax with random image transformations

Due to the randomness of the transforms, samples of the transformed image will have different softmax outputs. With each image, we obtain a distribution over the softmax outputs accumulated from multiple samples of the transformation. These are the steps to obtain the distribution of softmax:

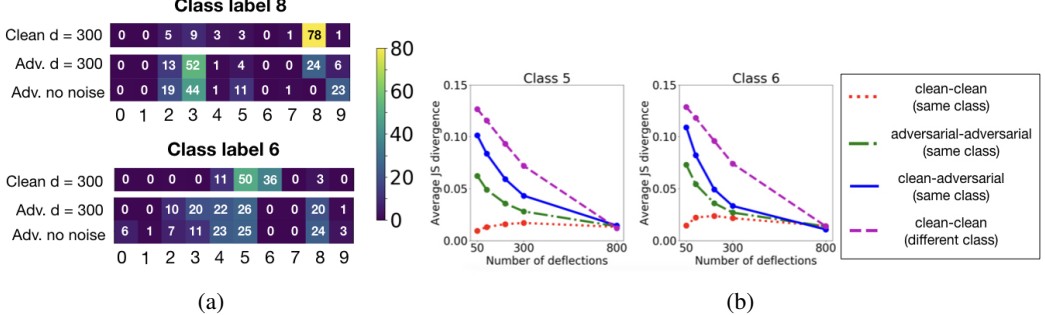

(a)  (b)

Figure 2: MNIST dataset: (a) Effect of number of pixel deflections ($d$) on the voting predictions of clean and adversarial images for MNIST digits 6 and 8. The numbers represent the percentage of images predicted to the various classes, eg. for class label 8, 78% of the transformed clean images are predicted correctly. For class label 8, with the image transformation, there is recovery on the adversarial images (24% recovery), but on the clean images, there is a deterioration in performance as some images are misclassified. However, it is interesting that the misclassifications have the same voting classes as the transformed adversarial images. Similar overlap in voting classes is observed for digit 6. (b) Effect of increasing number of pixel deflections on the distances of distributions obtained from clean and adversarial images. The standard error bars, taken over 3 random seeds for the transformation, are smaller than the plot points. Best viewed in color.

1. For each input image $x$, obtain $N$ transformed samples: $t_x^{(i)} \sim T(x), i = 1, \cdots, N$

2. The transformed samples of the image $(t_x^{(1)}, t_x^{(2)}, \cdots, t_x^{(N)})$ are fed into the CNN individually to obtain their softmax probabilities. Let $\sigma_x^{(i)}$ be the softmax vector derived from $t_x^{(i)}$, and $\sigma_{x,j}^{(i)}$, for $j = 1, \cdots, C$, be the $j$-th component of the softmax vector. $C$ denotes the number of classes for the classification task. With each input image and a transformation method, there exists an underlying joint distribution of the CNN softmax probabilities, from which we estimate with $N$ samples.

3. The underlying joint distribution of the softmax has a dimension equal to the number of classes (eg. 10-D for MNIST). Performing accurate density estimation in high dimensions is challenging due to the curse of dimensionality. Here we make an approximation by computing the marginal distributions over each class. When we use the term 'distribution of softmax', we are referring to the marginalized distributions. We use kernel density estimation with a Gaussian kernel. Let $\mathbf{h}_{x,j}$ be the distribution accumulated from $\sigma_{x,j}^{(1)}, \cdots, \sigma_{x,j}^{(N)}$:

$$h_{x,j}(s) = \frac{1}{N\sqrt{2\pi}\delta} \sum_i^N \exp\left( -\frac{(s - \sigma_{x,j}^{(i)})^2}{2\delta^2} \right),  \quad (1)$$

where $\delta$ is the kernel width and $s \in [0, 1]$ is the support of the softmax output. The distribution is then discretized into bins.

In this section, we study the effect of image transformation on the distribution of the softmax and make several interesting observations. In the following analyses, we study a LeNet5 CNN (LeCun et al., 1998) trained with MNIST. The adversarial images are generated using FGSM and for the transformation defense, we use pixel deflection, with $N$=100 transformation samples per image. The image transformation magnitude is controlled by the number of pixel deflections, $d$. In the analysis here and in the experimental results in Section 5, when reporting the accuracies, on clean images, we consider images that have been correctly predicted by the CNN, hence without any transformation defense, the test accuracy is 100%. This is following the setup of Prakash et al. (2018) where the misclassified images by CNN are excluded as it is not meaningful to evaluate any attack (and subsequent defense) methods on these images. For adversarial images, we consider the images that have been successfully attacked, so the test accuracy reflects the recovery rate and without any defense the accuracy is 0%.

In Figure 2a, we show how the image transformation affects the voting predictions on two MNIST classes. For each MNIST class, we take all the clean and adversarial test images, perform the transformation and then feed through the CNN to obtain the final voting prediction. We observe that for class label 8, there is some recovery from the attack as some adversarial images are voted to the correct class after the transformation. However, some clean images get misclassified to other classes (eg. 2 and 3). Although this means there is a deterioration of the accuracy on clean images, it is interesting that the misclassifications have the same voting classes as the transformed adversarial images. A similar pattern is observed for class label 6 where the clean images are misclassified to classes 4 and 5, which overlap with the vote predictions of some adversarial images at $d = 300$.

With the above analysis, we characterize the relationship between the clean and adversarial images in terms of the JS divergence of the distributions of the softmax at increasing number of pixel deflections. For each MNIST digit class, we quantify the (1) distance of the distributions among the clean images (clean-clean, same class), (2) distance of the distributions among the adversarial images (adversarial-adversarial, same class), (3) the distance of the distributions between clean and adversarial images (clean-adversarial, same class) and (4) the distance of the distributions between clean images of this class and all other classes (clean-clean, different class). Here we give details on the calculation of the 4 distance measures. First, the distance between the distributions of softmax output for two input images, $x_1$ and $x_2$ is given by $d(h_{x_1}, h_{x_2}) = \frac{1}{C} \sum_j^C D_{JS}(h_{x_1,j}, h_{x_2,j})$, where $D_{JS}$ is the Jensen-Shannon divergence. Distance measures of (1) and (2) computed by taking the average distance of each image distribution to the centroid distribution which is computed with $\mu(\{\mathbf{h}_{x_1,j}, \cdots, \mathbf{h}_{x_M,j}\}) = \frac{1}{M} \sum_i^M \mathbf{h}_{x_i,j}$. (3) is computed by the distance between the centroids of the clean and adversarial distributions. Finally, (4) is computed by the distance of the centroid distribution of the clean images of the particular class with the centroid distribution of another class, averaged over the other 9 classes.

In Figure 2b, we show results for two MNIST classes, but similar trends are observed across all classes (see Figure 8 in Appendix A)). The clean-clean (same-class) distance starts off low initially as all clean samples will give high scores at the correct class. With increasing number of deflections, there is increased variability in the softmax outputs and the resulting distributions. Next, the adversarial images of the same class are initially predicted as different incorrect classes without any transformation, and hence the adversarial-adversarial (same-class) distance starts off high and decreases with more transformation. The clean-adversarial (same-class) distance decreases with increasing image transformation which shows that the distributions of softmax from the clean and adversarial images are becoming more similar. Finally, the clean-clean (different class) distance decreases as well, which is expected because we already know that with more transformation, the clean image voting accuracy deteriorates. However, we observe that clean-clean (different class) distance decreases less rapidly and remains higher than clean-clean (same-class) distance at $d$=300. This means the transformation still retains information about the differences between the classes. At $d$=800, all 4 distance measures converge, which suggests the number of deflections is too large and the differences between the classes are no longer retained.

Next, we visualize the morphing of the distributions with increasing number of pixel deflections for an example image in Figure 3. For the purpose of visualization, instead of the per-class marginal distributions of the softmax, we perform kernel density estimation (kde) on the softmax values for the marginals on class 5 and 6. The softmax values of the other 8 classes are not shown. We have not excluded the areas where performing kde results in sum probability exceeding one, and our visualization still conveys our ideas and the distribution shapes well. Without any image transformation, as expected, the softmax outputs of the clean and adversarial images are very different. As the number of pixel deflections increases, each point evolves to a distribution due to the randomness of the transformation. The voting mechanism is straightforward; an image is classified to the class where the distribution mass is largest. In this example, the distribution shapes for the clean and adversarial image become more similar, and result in the same incorrect voting prediction at $d$=300. This shows the similarity of distributions obtained from clean and adversarial images after image transformation, which was illustrated in Figure 2b.

In Figure 4, we show more examples of the distributions obtained from clean images ($A$-$H$) and their adversarial counterparts ($\tilde{A}$-$\tilde{H}$) at $d$=300. For clean images $A$-$D$, voting predicts correctly but on the adversarial counterparts $\tilde{A}$-$\tilde{D}$, voting predicts wrongly. For clean images $E$-$H$ and the adversarial counterparts $\tilde{E}$-$\tilde{H}$, voting predicts wrongly. For completeness, we also show in Figure 9

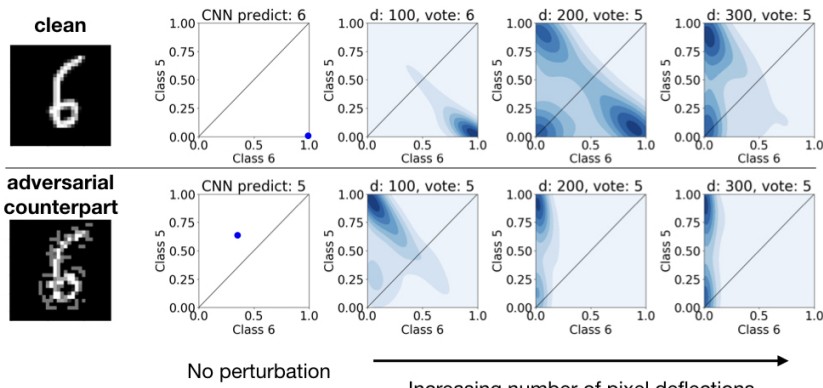

Figure 3: An example image and its adversarial counterpart (ground truth: class 6) undergoing increasing number of pixel deflections ($d$). The clean image gets misclassified by voting as the distribution mass shifts to the incorrect class. Note that for the purpose of visualization, instead of the per-class marginal distributions of the softmax, here we show the joint distribution over the softmax values of classes 5 and 6. The softmax values of the other 8 classes are not shown.

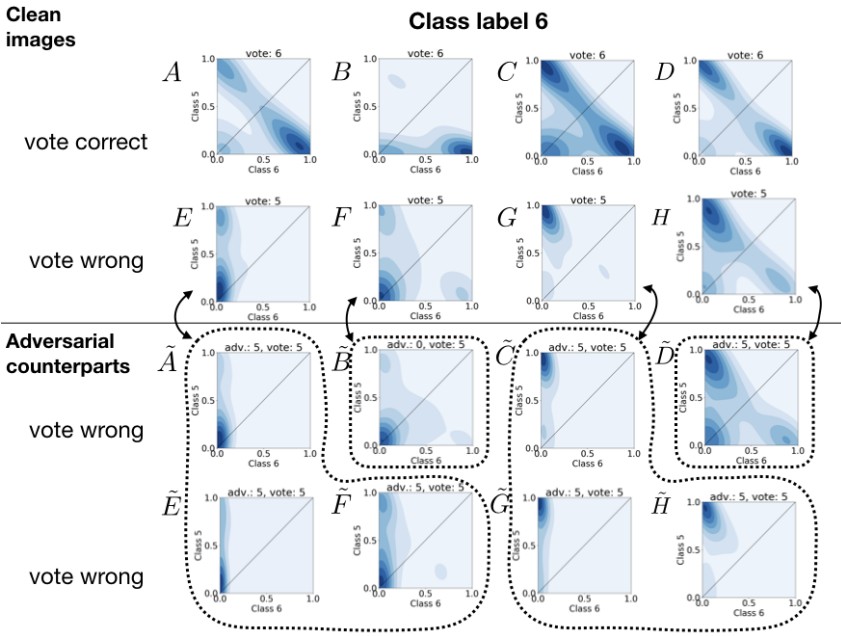

Figure 4: Distributions of the softmax obtained from 8 selected clean images of MNIST digit 6 ($A$ -$H$) and their adversarial counterparts ($\tilde{A}$-$\tilde{H}$) at $d$=300. Here we show the joint distribution of the softmax at class 5 and 6. The clean and adversarial images misclassified by voting show similar distribution shapes, as indicated by the groupings and arrows, eg. between $E$ and $\tilde{A}, \tilde{E}, \tilde{F}$, between $F$ and $\tilde{B}$ and so on.

in Appendix B examples of adversarial images where the transformation defense, coupled with voting, has successfully recovered the correct class. With the random image transformation, there are similarities in the distribution shapes between the clean and adversarial images, as shown by the groupings and arrows (eg. between $E$ and $\tilde{A}, \tilde{E}, \tilde{F}$). This further supports our earlier observations. After the image transformation, the voting accuracy on the clean images deteriorates, but the resulting distributions have similar features as the distributions from the adversarial counterparts. This gives us an idea to enhance existing transformation-based defenses: to train a distribution classifier on the distributions obtained from clean images only, while improving the performance on both clean and adversarial images.

## 4 ENHANCING TRANSFORMATION-BASED DEFENSES WITH DISTRIBUTION CLASSIFIER

Instead of voting, to reduce the drop in performance on clean images, we train a separate compact distribution classifier to recognize patterns in the distributions of softmax probabilities of clean images, as illustrated in Figure 1. For each clean image, the marginal distributions obtained are inputs to the distribution classifier, which learns to associate this distribution with the correct class label. If the individual transformed images were initially misclassified by the CNN, our distribution classifier should learn to recover the correct class. During the test phase, for any input image, clean or adversarial, we build the distribution of softmax from $N$ transformed samples and feed them into our trained distribution classifier to obtain the final prediction. Note that our defense method does not require retraining of the original CNN, is agnostic to the attack method and can be integrated with most existing stochastic transformation-based methods.

**Distribution classifiers:** We investigate three distribution classification methods. First, we adapt a state-of-the-art distribution-to-distribution regression method, called distribution regression network (DRN) (Kou et al., 2018) (details are included in Appendix C). We also experimented on random forest (RF), which averages the outputs from multiple decision trees. Finally, we experimented on multilayer perceptrons (MLP) which are fully connected neural networks, with a softmax output layer. For this distribution classification task, we concatenate the distribution bins from the softmax classes into a single input vector for RF and MLP. For DRN and MLP, we use the cross entropy loss and the network architectures are chosen by cross-validation. For random forest, the Gini impurity is used as the splitting criterion and the number of trees and maximum depth are tuned by cross-validation. The hyperparameter values are included in Appendix D.4.

## 5 EXPERIMENTS AND DISCUSSION

In the following section, we describe our experimental setup to evaluate the performance on clean and adversarial images with our distribution classifier method.

**Datasets and CNN networks:** We use the MNIST (LeCun et al., 1998), CIFAR10 and CIFAR100 (Krizhevsky & Hinton, 2009) datasets. For the CNN model for MNIST, we use LeNet5 (LeCun et al., 1998) that has 98.7% test accuracy. For CIFAR10 and CIFAR100, we use wide ResNet (Zagoruyko & Komodakis, 2016) with test accuracies of 95.7% and 78.9% respectively.

**Attack methods:** As introduced in Section 2, we use four adversarial attacks in the untargeted setting. In Appendix D.1, we have included the distortion metrics, the success rates and the hyperparameters. The attacks are implemented using the CleverHans library (Papernot et al., 2018).

**Transformation-based defenses:** As a baseline, we use a random pixel noise (RPN) as a defense method, where each pixel noise is sampled with a uniform distribution with $L_\infty$ measure. In addition, we use two existing transformation-based methods: pixel deflection (PD) (Prakash et al., 2018) and image random resize and pad (RRP) (Xie et al., 2017). Although these two methods have not been tested for MNIST, CIFAR10 and CIFAR100, we find that they work considerably well and present the results here. The hyperparameter tuning for each defense is conducted on the validation sets. We select hyperparameters that give the best recovery from adversarial attack, regardless of the deterioration in accuracy on clean images. The hyperparameters are included in Appendix D.2.

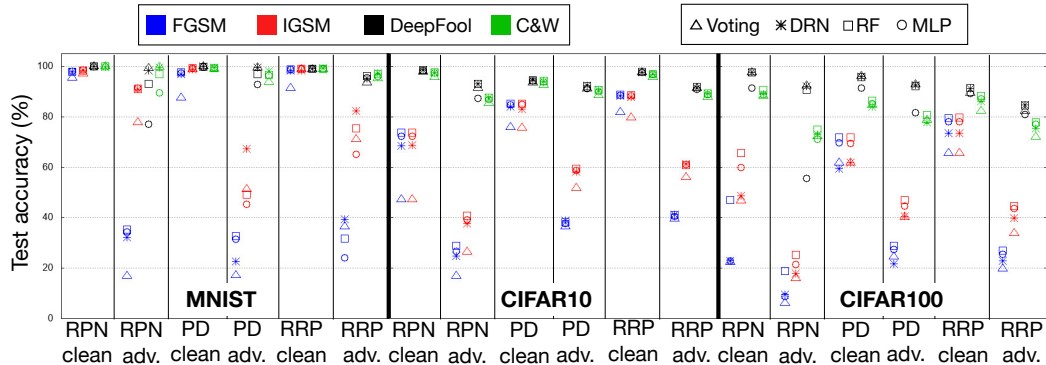

Figure 5: MNIST, CIFAR10 and CIFAR100 results: For each attack and transformation-based defense, we compare the clean and adversarial (adv.) test accuracies with baseline majority voting and the three distribution classifier methods. The distribution classifiers generally show improvements over voting. Best viewed in color.

To test the effectiveness of the transformation-based defenses before integrating with our defense method, we perform majority voting on the transformed image samples. This sets the baseline for our distribution classifier defense method. When reporting the test accuracies, on clean images, we consider images that have been correctly predicted by the CNN, hence without any defense method, the test accuracy is 100%. For adversarial images, we consider the images that have been successfully attacked, so the test accuracy reflects the recovery rate and without any defense the accuracy is 0%.

## 5.1 MNIST RESULTS

For the MNIST dataset, $N = 100$ transformation samples were used for voting and for constructing the distribution of softmax. We found that the distribution classifiers required only 1000 training data, which is a small fraction out of the original 50,000 data. Figure 5 (left) shows the test accuracies of the three transformation-based defenses with majority voting and with the three distribution classifiers. Table 11 in Appendix D.5 shows the numerical figures of the results. First, we observe that the recovery on adversarial images with majority voting for the iterative methods IGSM, Deep-Fool and C&W is much better compared to single-step FGSM. This is in line with the observations by Xie et al. (2017) where they found their defense to be more effective for iterative attacks.

The distribution classifiers have improved accuracy over voting on the clean images, except when the voting accuracy was already high (eg. 100% voting accuracy for PD on DeepFool). The mean improvement of the accuracy on the clean images is 1.7% for DRN. Hence, our distribution classifier method is stronger than voting. Voting simply takes the mode of the softmax probabilities of the transformed image, disregarding properties such as variance across the classes. In contrast, the distribution classifier learns from the distinctive features of the distribution of softmax.

Without training on the distributions obtained from adversarial images, our method has managed to improve the recovery rate, with a mean improvement of 5.9% for DRN. The three distribution classifier methods are comparable, except for some cases where DRN outperforms other classifiers (eg. PD adv., IGSM) and where MLP and RF have lower accuracy than voting (eg. RPN adv., DeepFool and C&W). In Figure 4, we show that after image transformation, the distributions of softmax between the clean and adversarial images show some similarities and distinctive features. In fact, all of the clean ($A$-$H$) and adversarial ($\tilde{A}$-$\tilde{H}$) images (class 6) are classified correctly by the distribution classifier. Even though the distribution classifier was only trained on distributions from the clean images ($A$-$H$), the distribution classifier can recover the correct class for the adversarial images where voting has failed ($\tilde{A}$-$\tilde{H}$). The distribution classifier does so by learning the distinctive shapes of the distributions associated with the digit class from the clean images, and is able to apply this to the adversarial images with similar distribution shapes. Furthermore, our distribution classifier is able to pick up subtle differences in the distribution features. Figure 6a shows examples of clean images with class label 5 that are correctly classified by our distribution classifier. It is

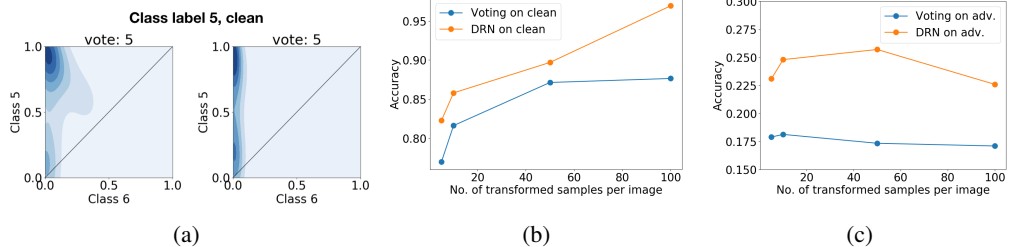

Figure 6: MNIST dataset: (a) Example of two clean images with class label 5, where the distribution shapes look similar to $\tilde{C}$ and $G$ in Figure 4 but the distribution classifier can still discriminate between the distributions from class labels 5 and 6 and classify them correctly. (b) On the clean images, both voting and DRN accuracies improve with more number of transformed samples. As number of samples increases, voting saturates while DRN continues to improve. (c) On the adversarial images, the accuracies stay more of less the same.

interesting that although the distribution shapes for adversarial images $\tilde{C}$ and $G$ shown in Figure 4 look similar, our distribution classifier is able to distinguish between the shapes for class 5 and 6.

### 5.1.1 NUMBER OF TRANSFORMED SAMPLES REQUIRED

We used $N$=100 transformed samples in our experiments. Hence, the evaluation time will be 100 times longer than taking a single sample. Here we study the effect of the number of samples. Figure 6b and 6c show the classification accuracies for voting and DRN as the number of transformed samples increases. On the clean images, both voting and DRN accuracies improve with more number of samples, with the performance of voting saturating while DRN's performance continues to increase with widening gap. This shows that a sufficient number of samples is required to capture the features of the distribution of softmax. On the adversarial images, the accuracies stay more of less the same. Although having more transformed samples is beneficial for the performance on clean images, our distribution classifier improves the voting performance regardless of the number of samples.

### 5.2 CIFAR10 AND CIFAR100 RESULTS

For the CIFAR10 and CIFAR100 datasets, $N = 50$ image transformation samples and 10,000 training data were used. Figure 5 (middle) shows the results for CIFAR10. All three distribution classifiers gave comparable improvements over voting, except for MLP which performs worse than voting for adversarial images with RPN on DeepFool. For CIFAR100 (Figure 5, right), the distribution classifiers mostly show improved performance over voting. There are exceptions where DRN (eg. PD adv., FGSM) and MLP (eg. RPN adv., DeepFool) have lower accuracy than voting. This suggests that for datasets with more classes, random forest may perform better than other classifiers.

As explained in Section 3, in the results in Figure 5, we have excluded clean images which are misclassified by the CNN and the images where the attack has failed. To check that our method works on these images, we evaluated these images for CIFAR100 with FGSM attack, random resize and padding and random forest classifier. Our results in Table 14 in the Appendix show that our distribution classifier method still outperforms majority voting.

## 6 END-TO-END ATTACK ON DISTRIBUTION CLASSIFIER METHOD

Here we evaluate end-to-end attacks on our distribution classifier method (with DRN) on MNIST and CIFAR10. We use Boundary Attack (Brendel et al., 2017) which is a black-box decision-based attack. We performed the attack on the base CNN classifier (CNN), CNN with pixel deflection and voting (Vote), and CNN with pixel deflection and distribution classifier trained on clean images (DRN). In addition, we trained the distribution classifier on a mixture of distributions obtained from both clean and adversarial images obtained with IGSM on the base CNN, which can be seen as a lightweight adversarial training (DRN LAT) except that the CNN is kept fixed. Finally, we tested the attack on an adversarially-trained CNN (Adv trained CNN) by Madry et al. (2017) with allowed

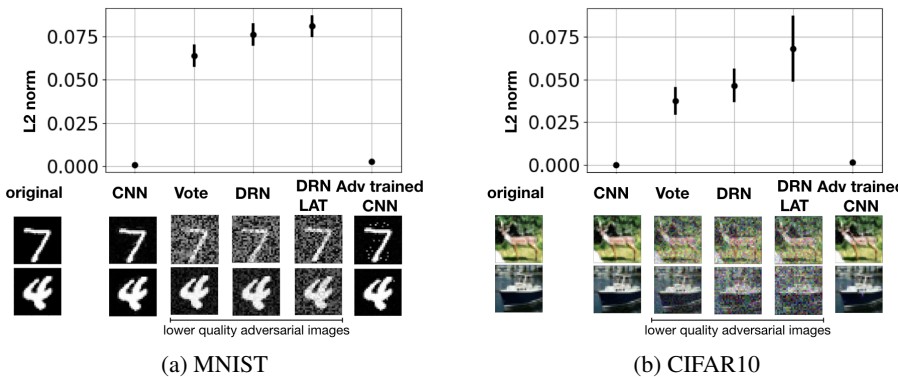

(a) MNIST          (b) CIFAR10

Figure 7: Black-box boundary attack on MNIST and CIFAR10, with the average $L_2$ of the attack perturbation shown, with the bars denoting the standard error of the means. Note that the adversarial images for vote, DRN and DRN LAT are of lower quality.

perturbations of $L_\infty \leq 0.3$. Since Boundary Attack uses the $L_2$ measure, the adversarially-trained CNN which uses the $L_\infty$ metric is not expected to perform well. For details of our implementation of Boundary Attack, please refer to Appendix E. Figure 7 shows the mean $L_2$ of the perturbations over 100 test images, with a maximum of 5000 iterations for the attack. CNN and the adversarially-trained CNN have very low perturbations. The stochastic models, Vote, DRN and DRN LAT, have much higher perturbations with lower quality adversarial images, and the difficulty of the attack increases in that order. This shows that the distribution classifier and the lightweight adversarial training extension are more difficult to attack by the Boundary Attack method compared to voting.

Athalye et al. (2018) have shown that under the white-box setting where the attacker has full knowledge of the CNN model and the defense, random transformation defenses are susceptible to further attack by estimating the gradients using multiple transformation samples, in a method called Expectation over Transformation(EOT). To employ white-box attack on our distribution classifier method, there are a few potential challenges. First, we use 50 to 100 transformed samples per image to accumulate the distribution of softmax. Attacking our method with EOT will be very time-consuming as it requires taking multiple batches of transformations, each with 50-100 samples. Next, we have shown our method works with different distribution classifier models, including the non-differentiable random forest. While there have been attacks proposed for random forests (Kantchelian et al., 2016), it is unclear how feasible it is to combine these attacks with EOT. We leave the evaluation of white-box attacks on our distribution classifier method for future work.

## 7 CONCLUSION

Adversarial attacks on convolutional neural networks have gained significant research attention and stochastic input transformation defenses have been proposed. However, with transformation-based defenses, the performance on clean images deteriorates and the exact mechanism in which how this happens is unclear. In this paper, we conduct in-depth analysis on the effects of stochastic transformation-based defenses on the softmax outputs of clean and adversarial images. We observe that after image transformation, the distributions of softmax obtained from clean and adversarial images share similar distinct features. Exploiting this property, we propose a method that trains a distribution classifier on the distributions of the softmax outputs of transformed clean images only, but show improvements in both clean and adversarial images over majority voting. In our current work, we have considered untargeted attacks on the CNN and it is interesting to test our distribution classifier method with targeted attacks.

### ACKNOWLEDGMENTS

We thank Harold Soh, Wang Wei, Terence Sim, Mahsa Paknezhad and Kaicheng Liang for their constructive discussions. This work is supported, in part, by the Biomedical Research Council of the Agency for Science, Technology and Research and the National University of Singapore.

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

## A DISTANCE BETWEEN DISTRIBUTIONS OF SOFTMAX FOR ALL MNIST CLASSES

In Section 3, we studied the 4 distance metrics for the distribution of softmax. Figure 8 shows the distance metrics for all ten MNIST classes with increasing number of pixel deflections.

## B EXAMPLES OF VOTING RECOVERING FROM ADVERSARIAL ATTACK

In Figure 9, we show examples where pixel deflection with voting recovers from the adversarial attack.

## C ADAPTATION OF DRN FOR DISTRIBUTION CLASSIFICATION

For one of the distribution classifier methods, we adapt a state-of-the-art distribution-to-distribution regression method, called distribution regression network (DRN) (Kou et al., 2018). DRN encodes an entire distribution in each network node and this compact representation allows it to achieve higher prediction accuracies for the distribution regression task compared to conventional neural networks. Since DRN shows superior regression performance, we adapt DRN for distribution classification in this work.

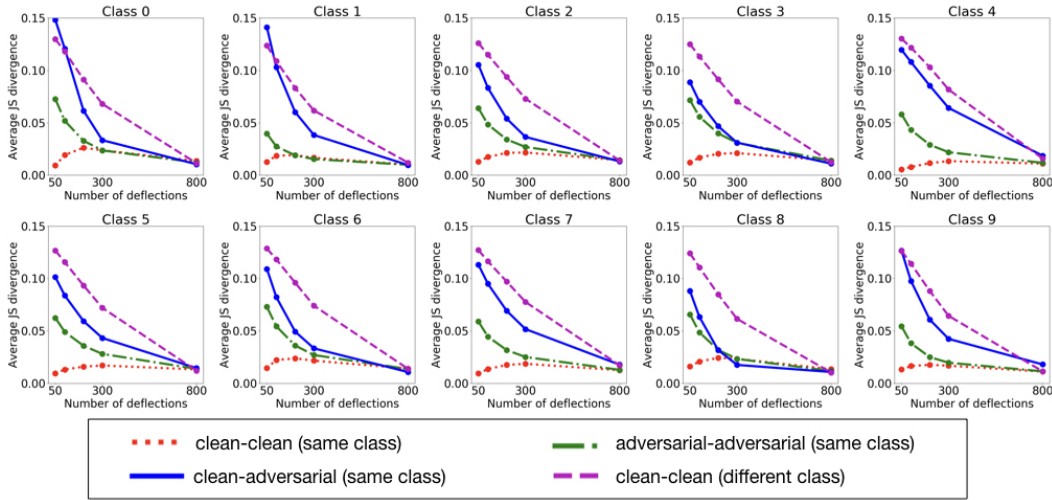

Figure 8: Effect of increasing number of pixel deflections on the distances of distributions obtained from clean and adversarial images of the MNIST dataset. Best viewed in color.

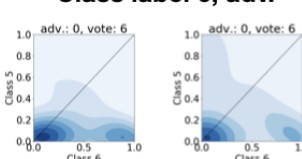

Figure 9: Adversarial examples where pixel deflection with voting recovers from adversarial attack, where ground truth label is 6 but CNN predicts as 0.

Our adaption of the distribution classifier is shown on the right of Figure 10. The network consists of fully-connected layers, where each node encodes a distribution. The number of hidden layers and nodes per hidden layer are chosen by cross validation. The number of discretization bins for each distribution for the input layer and hidden layers is also tuned as hyperparameters. To adapt DRN for our distribution classification task, for the final layer, we have $C$ nodes representing each class and we use 2 bins for each distribution to represent the logit output for the corresponding class. The cost function for the distribution classifier is the cross entropy loss on the logits. The distribution classifier is optimized by backpropagation using the Adam optimizer (Kingma & Ba, 2014). The weight initialization method follows Kou et al. (2018), where the weights are sampled from a uniform random distribution.

## D  EXPERIMENTAL SETUP

### D.1  ADVERSARIAL ATTACK HYPERPARAMETERS

Tables 1 to 3 show the hyperparameter settings used for the adversarial attacks. The attacks are implemented using the CleverHans library (Papernot et al., 2018). For DeepFool and C&W, the other hyperparameters used are the default values set in CleverHans. For $L_2$ norm, we use the root-mean-square distortion normalized by total number of pixels, following previous works.

### D.2  IMAGE TRANSFORMATION DEFENSE PARAMETERS

Tables 4 to 6 show the image transformation parameters used for MNIST and CIFAR10 respectively. The hyperparameter tuning for each defense method is conducted on the validation set for each

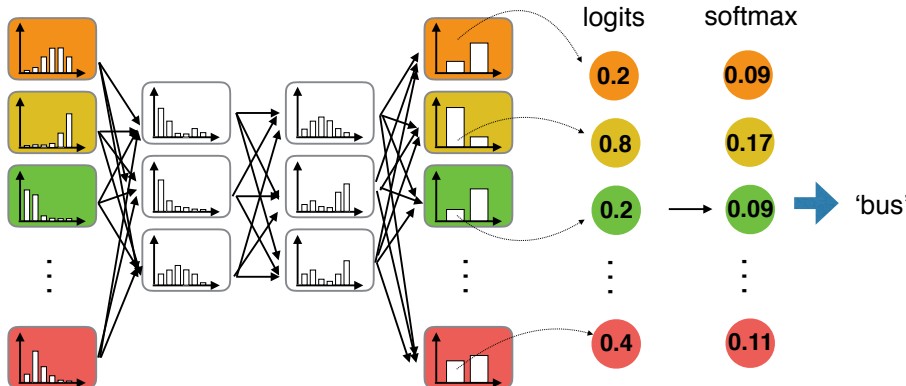

Figure 10: Adaptation of DRN for the distribution classifier task.

Table 1: Hyperparameter settings for the four adversarial attack methods on the MNIST dataset.

| | | MNIST | |
| --- | --- | --- | --- |
| | Distortion | Attack success rate (%) | Settings |
| FGSM | $L_\infty = 0.39$ | 53.2 | - |
| IGSM | $L_\infty = 0.24$ | 91.2 | 100 steps, step size=0.004 |
| DeepFool | $L_2 = 0.06$ | 97.6 | Max iter.=30 |
| C&W | $L_2 = 0.142$ | 94.9 | Max iter.=500 |

Table 2: Hyperparameter settings for the four adversarial attack methods on the CIFAR10 dataset.

| | | CIFAR10 | |
| --- | --- | --- | --- |
| | Distortion | Attack success rate (%) | Settings |
| FGSM | $L_\infty = 0.031$ | 58.2 | - |
| IGSM | $L_\infty = 0.031$ | 100.0 | 20 steps, step size=0.004 |
| DeepFool | $L_2 = 0.042$ | 95.3 | Max iter.=5 |
| C&W | $L_2 = 0.023$ | 94.1 | Max iter.=10 |

dataset. We select hyperparameters that give the best recovery from adversarial attack, regardless of the deterioration in accuracy on clean images.

### D.3 CLASS ACTIVATION MAPS FOR PIXEL DEFLECTION

The pixel deflection (Prakash et al., 2018) defense uses class activation maps (CAMs) (Zhou et al., 2016) to randomly select pixels to undergo the deflection step. In our experiments, we did not use class activation maps and instead randomly select pixels with equal probabilities. First, for the MNIST dataset, CAMs are unsuitable because the LeNet (LeCun et al., 1998) architecture does not have global average pooling layers which are required for CAMs. For the CIFAR10 dataset, the wide ResNet (Zagoruyko & Komodakis, 2016) architecture uses a final layer of global average pooling and so we tested CAMs on it. Table 7 compares the performance on clean and adversarial images using the FGSM and IGSM attacks, with and without CAMs, which shows that using CAMs does

Table 3: Hyperparameter settings for the four adversarial attack methods on the CIFAR100 dataset.

| | CIFAR100 | | |
|---|---|---|---|
| | Distortion | Attack success rate (%) | Settings |
| FGSM | $L_\infty = 0.031$ | 78.5 | - |
| IGSM | $L_\infty = 0.031$ | 99.9 | 20 steps, step size=0.004 |
| DeepFool | $L_2 = 0.051$ | 91.8 | Max iter.=5 |
| C&W | $L_2 = 0.026$ | 96.0 | Max iter.=10 |

Table 4: Details of the image transformation parameters for MNIST. The three transformation-based methods tested are random pixel noise (RPN), pixel deflection (PD) and random resize and padding (RRP). For RPN, the noise magnitude is unnormalized (out of 255). For PD, d is the number of deflections, w is the window size and $\sigma$ is the denoising parameter.

| | FGSM | IGSM | DeepFool | C&W |
|---|---|---|---|---|
| RPN | $L_\infty$=130 | $L_\infty$=90 | $L_\infty$=10 | $L_\infty$=10 |
| PD | d=300, w=20, $\sigma$=0 | d=100, w=20, $\sigma$=0 | d=10, w=20, $\sigma$=0.08 | d=100, w=25, $\sigma$=0.08 |
| RRP | resize range=[4,28] | resize range=[22,28] | resize range=[23,28] | resize range=[23,28] |

not cause significant difference in performance. This may be because CAMs are more effective on larger images such as those in ImageNet where there are many more background pixels.

### D.4    DISTRIBUTION CLASSIFIER HYPERPARAMETERS

Our defense method uses a distribution classifier to train on distributions of softmax probabilities obtained from transformed samples of the clean images. For each image, we build the marginal distributions of the softmax for each class using kernel density estimation with a Gaussian kernel. The kernel width is optimized to be 0.05. For DRN and MLP, the network architecture of the distribution classifier and optimization hyperparameters are chosen by cross-validation. For random forest, the number of trees and maximum depth of the trees are tuned by cross-validation. The hyperparameters used are shown in Tables 8 to 10.

### D.5    ACCURACY RESULTS FOR DISTRIBUTION CLASSIFIERS

Here we include the detailed numerical figures for the accuracies of majority voting and the distribution classifier methods. Tables 11 to 13 show the clean and adversarial test accuracies and the 4 attack methods and the 3 defense methods.

## E    DETAILS OF IMPLEMENTATION OF BOUNDARY ATTACK

For Vote, DRN and DRN LAT, the model outputs are random because of the random image transformation. At each step of Boundary Attack, we allow the attack to query the model once, and this involves taking 50-100 transformed samples for the image to perform voting or to feed to the

Table 5: Details of the image transformation parameters for CIFAR10.

| | FGSM | IGSM | DeepFool | C&W |
|---|---|---|---|---|
| RPN | $L_\infty$=40 | $L_\infty$=40 | $L_\infty$=7 | $L_\infty$=10 |
| PD | d=60, w=10, $\sigma$=0.06 | d=80, w=10, $\sigma$=0.06 | d=20, w=10, $\sigma$=0.04 | d=80, w=10, $\sigma$=0.04 |
| RRP | resize range=[20,25] | resize range=[19,25] | resize range=[28,32] | resize range=[23,32] |

Table 6: Details of the image transformation parameters for CIFAR100.

|  | FGSM | IGSM | DeepFool | C&W |
|---|---|---|---|---|
| RPN | $L_\infty$=40 | $L_\infty$=25 | $L_\infty$=3 | $L_\infty$=9 |
| PD | d=90, w=10, $\sigma$=0.06 | d=80, w=10, $\sigma$=0.06 | d=20, w=10, $\sigma$=0.02 | d=30, w=10, $\sigma$=0.04 |
| RRP | resize range=[23,26] | resize range=[23,26] | resize range=[27,32] | resize range=[25,30] |

Table 7: Performance for pixel deflection on the CIFAR10 dataset with and without class activation maps (CAM). Numbers shown are the test accuracies (%).

|  | FGSM | | IGSM | |
|---|---|---|---|---|
|  | Without CAM | With CAM | Without CAM | With CAM |
| clean | 75.96 (0.07) | 75.98 (0.02) | 75.71 (0.04) | 75.70 (0.06) |
| adv. | 36.35 (0.07) | 36.40 (0.02) | 51.66 (0.07) | 51.59 (0.08) |

Table 8: DRN network architecture for the distribution classifier, where 2x10 represents 2 hidden layers of 10 nodes.

|  |  | FGSM | IGSM | DeepFool | C&W |
|---|---|---|---|---|---|
|  | RPN | 1x10 | 2x10 | 1x10 | 1x10 |
| MNIST | PD | 1x10 | 1x10 | 1x20 | 1x20 |
|  | RRP | 1x10 | 1x10 | 1x20 | 1x20 |
|  | RPN | 1x10 | 1x20 | 1x20 | 1x20 |
| CIFAR10 | PD | 1x20 | 1x10 | 1x20 | 1x10 |
|  | RRP | 1x20 | 1x10 | 1x20 | 1x10 |
|  | RPN | 1x100 | 1x100 | 1x100 | 1x100 |
| CIFAR100 | PD | 1x100 | 1x100 | 1x100 | 1x100 |
|  | RRP | 1x100 | 1x100 | 1x100 | 1x100 |

Table 9: MLP network architecture for the distribution classifier, where 2x10 represents 2 hidden layers of 10 nodes.

|  |  | FGSM | IGSM | DeepFool | C&W |
|---|---|---|---|---|---|
|  | RPN | 1x20 | 1x20 | 1x50 | 1x20 |
| MNIST | PD | 1x20 | 1x50 | 1x50 | 1x50 |
|  | RRP | 1x50 | 1x20 | 1x50 | 1x50 |
|  | RPN | 1x50 | 1x50 | 1x50 | 1x20 |
| CIFAR10 | PD | 1x50 | 1x50 | 1x50 | 1x50 |
|  | RRP | 1x50 | 1x50 | 1x20 | 1x20 |
|  | RPN | 1x2000 | 1x1000 | 1x500 | 1x500 |
| CIFAR100 | PD | 1x500 | 1x1000 | 2x100 | 1x1000 |
|  | RRP | 1x200 | 1x1000 | 2x100 | 1x1000 |

Table 10: Random forest hyperparameters for the distribution classifier, where $n$ represents the number of trees and $d$ represents the maximum depth of the trees.

|  |  | FGSM | IGSM | DeepFool | C&W |
|---|---|---|---|---|---|
| MNIST | RPN | n=100, d=20 | n=100, d=20 | n=200, d=50 | n=200, d=50 |
|  | PD | n=100, d=20 | n=100, d=20 | n=200, d=50 | n=200, d=50 |
|  | RRP | n=200, d=20 | n=100, d=20 | n=200, d=50 | n=200, d=50 |
| CIFAR10 | RPN | n=200, d=50 | n=200, d=50 | n=200, d=50 | n=200, d=50 |
|  | PD | n=200, d=50 | n=200, d=50 | n=200, d=50 | n=200, d=50 |
|  | RRP | n=200, d=50 | n=200, d=50 | n=200, d=50 | n=200, d=50 |
| CIFAR100 | RPN | n=200, d=200 | n=200, d=200 | n=200, d=200 | n=200, d=200 |
|  | PD | n=200, d=200 | n=200, d=200 | n=200, d=100 | n=200, d=200 |
|  | RRP | n=200, d=200 | n=200, d=200 | n=200, d=100 | n=200, d=100 |

Table 11: MNIST results: For each attack, we compare the clean and adversarial (adv.) test accuracies with majority voting (Vote) and the three distribution classifier methods: distribution regression network (DRN), random forest (RF) and multilayer perceptron (MLP). The three transformation-based defenses are random pixel noise (RPN), pixel deflection (PD) and random resize and padding (RRP). With no defense, the clean accuracy is 100% and the adversarial accuracy is 0%.

| MNIST test accuracies (%) | | | | | | | |
|---|---|---|---|---|---|---|---|
|  |  | RPN | | PD | | RRP | |
|  |  | clean | adv. | clean | adv. | clean | adv. |
| FGSM | Vote | 95.4 | 17.0 | 87.6 | 17.1 | 91.4 | 36.4 |
|  | DRN | 97.8 | 32.1 | 96.9 | 22.6 | 98.2 | 39.4 |
|  | RF | 97.8 | 35.2 | 97.5 | 32.8 | 98.7 | 31.7 |
|  | MLP | 98.1 | 34.4 | 97.3 | 31.6 | 98.8 | 24.0 |
| IGSM | Vote | 97.2 | 77.9 | 98.9 | 51.3 | 98.7 | 71.1 |
|  | DRN | 98.4 | 90.9 | 98.8 | 67.3 | 98.4 | 82.4 |
|  | RF | 98.4 | 91.1 | 99.3 | 49.1 | 99.0 | 75.4 |
|  | MLP | 98.5 | 91.4 | 99.3 | 45.4 | 99.0 | 65.2 |
| DeepFool | Vote | 100 | 99.3 | 100 | 99.4 | 98.9 | 93.6 |
|  | DRN | 99.9 | 98.3 | 99.9 | 99.4 | 99.1 | 95.2 |
|  | RF | 99.9 | 93.1 | 99.8 | 97.0 | 99.1 | 96.1 |
|  | MLP | 99.8 | 77.0 | 99.8 | 92.8 | 99.0 | 95.4 |
| C&W | Vote | 99.9 | 99.7 | 99.0 | 93.8 | 98.9 | 95.3 |
|  | DRN | 99.7 | 99.5 | 99.4 | 98.0 | 99.2 | 97.4 |
|  | RF | 99.9 | 96.8 | 99.3 | 96.6 | 99.1 | 97.0 |
|  | MLP | 99.7 | 89.6 | 99.4 | 96.3 | 99.0 | 96.7 |

Table 12: CIFAR10 results: For each attack, we compare the clean and adversarial (adv.) test accuracies with majority voting (Vote) and the three distribution classifier methods: distribution regression network (DRN), random forest (RF) and multilayer perceptron (MLP). The three transformation-based defenses are random pixel noise (RPN), pixel deflection (PD) and random resize and padding (RRP). With no defense, the clean accuracy is 100% and the adversarial accuracy is 0%.

| CIFAR10 test accuracies (%) | | | | | | | |
|---|---|---|---|---|---|---|---|
| | | RPN | | PD | | RRP | |
| | | clean | adv. | clean | adv. | clean | adv. |
| FGSM | Vote | 47.3 | 16.9 | 75.9 | 36.4 | 81.9 | 39.6 |
| | DRN | 68.4 | 24.8 | 84.1 | 39.0 | 88.4 | 41.2 |
| | RF | 73.8 | 28.8 | 85.2 | 38.3 | 88.7 | 41.1 |
| | MLP | 72.2 | 26.8 | 84.6 | 37.7 | 88.6 | 40.5 |
| IGSM | Vote | 47.3 | 26.5 | 75.7 | 51.7 | 79.8 | 56 |
| | DRN | 68.8 | 37.7 | 83.1 | 58.2 | 87.9 | 61.3 |
| | RF | 73.8 | 40.7 | 85.2 | 59.4 | 88.5 | 60.9 |
| | MLP | 72.2 | 39.2 | 84.9 | 58.7 | 88.4 | 61.0 |
| DeepFool | Vote | 97.8 | 91.4 | 93.5 | 91.1 | 97.6 | 91.3 |
| | DRN | 98.6 | 93.0 | 94.6 | 92.3 | 97.7 | 91.5 |
| | RF | 98.6 | 93.0 | 94.5 | 92.2 | 97.9 | 91.9 |
| | MLP | 98.3 | 87.3 | 94.2 | 91.1 | 97.7 | 91.0 |
| C&W | Vote | 95.8 | 85.5 | 92.5 | 88.5 | 95.8 | 87.7 |
| | DRN | 97.5 | 87.6 | 94.1 | 90.8 | 97.0 | 89.3 |
| | RF | 97.7 | 87.6 | 94.2 | 90.6 | 96.9 | 89.2 |
| | MLP | 97.3 | 86.8 | 93.9 | 89.9 | 96.8 | 88.8 |

Table 13: CIFAR100 results: For each attack, we compare the clean and adversarial (adv.) test accuracies with majority voting (Vote) and the three distribution classifier methods: distribution regression network (DRN), random forest (RF) and multilayer perceptron (MLP). The three transformation-based defenses are random pixel noise (RPN), pixel deflection (PD) and random resize and padding (RRP). With no defense, the clean accuracy is 100% and the adversarial accuracy is 0%.

| CIFAR100 test accuracies (%) | | | | | | | |
|---|---|---|---|---|---|---|---|
| | | RPN | | PD | | RRP | |
| | | clean | adv. | clean | adv. | clean | adv. |
| FGSM | Vote | 22.4 | 6.2 | 61.7 | 24.5 | 65.7 | 19.8 |
| | DRN | 22.8 | 9.5 | 59.5 | 21.8 | 73.5 | 23.0 |
| | RF | 46.9 | 18.9 | 71.8 | 28.9 | 79.5 | 27.0 |
| | MLP | 22.8 | 8.9 | 69.8 | 27.4 | 78.0 | 25.5 |
| IGSM | Vote | 46.7 | 16.1 | 61.9 | 40.3 | 65.7 | 34.0 |
| | DRN | 48.6 | 17.8 | 61.8 | 40.6 | 73.5 | 39.9 |
| | RF | 65.7 | 25.2 | 71.9 | 47.1 | 79.6 | 44.7 |
| | MLP | 59.8 | 21.4 | 69.4 | 44.6 | 78.1 | 43.7 |
| DeepFool | Vote | 97.7 | 92.4 | 96.1 | 92.8 | 89.4 | 82.3 |
| | DRN | 97.7 | 92.2 | 95.9 | 92.5 | 91.2 | 84.6 |
| | RF | 97.5 | 90.8 | 95.8 | 92.1 | 91.5 | 84.5 |
| | MLP | 91.4 | 55.6 | 91.3 | 81.7 | 89.2 | 80.8 |
| C&W | Vote | 88.3 | 72.3 | 84.6 | 78.8 | 82.4 | 72.0 |
| | DRN | 89.1 | 73.0 | 84.1 | 77.8 | 86.2 | 75.5 |
| | RF | 90.6 | 74.9 | 86.5 | 80.7 | 88.2 | 77.9 |
| | MLP | 88.8 | 71.0 | 85.1 | 78.4 | 87.1 | 77.1 |

Table 14: Distribution classifier outperforms majority voting for clean images that are misclassified by CNN and images where the attack has failed. The results are for CIFAR100 with FGSM attack, random resize and padding and random forest classifier.

|  | Test accuracy (%) | | |
| --- | --- | --- | --- |
|  | No transformation | Voting | Distribution classifier |
| Clean images misclassified by CNN | 0 | 12.3 | 18.1 |
| Images where attack has failed | 100 | 65.7 | 79.5 |

distribution classifier to obtain a prediction. To avoid overfitting to a fixed transformation pattern, the transformation is random at each step. Our criteria for an image being adversarial is that out of 5 queries, the image is misclassified at least once. Because of the randomness of the model, the image returned by Boundary Attack may be classified to the correct class, and we increase the perturbation by increasing amounts until the image is misclassified. Note that to overcome the randomness, we could have performed multiple queries at each attack step, but because our models already use 50-100 transformed samples per query, this will be computationally infeasible.

