# OpenReview forum: "Enhancing Transformation-Based Defenses Against Adversarial Attacks with a Distribution Classifier"
_ICLR.cc/2020/Conference — Accept (Poster)_

### Official Review · AnonReviewer1 · 2019-10-24
**Official Blind Review #1**

**Rating:** 6

**Review:**

This paper presents a novel defense method to make the classification more robust. The motivation is based on the observation: the distribution of the soft-max for the cleaned image and its transformed images for one class is similar to the distribution of the soft-max for the adversarial image and its transformed images for the same class, and the distributions of the soft-max for the cleaned image and its transformed images for different classes are different. Then, a distribution based method is proposed to classify the distribution of the soft-max for the cleaned (or adversarial) image and its transformed images.

After I read the core part several times, finally I understood the paper. Overall I think this paper is well motivated, and the empirical results also support the claims/observations. But I think this paper could be improved by (1) checking the performance over large datasets (such as ImageNet); (2) providing possible analysis on the observation. Otherwise, the readers cannot be fully convinced.

**Experience Assessment:**

I have read many papers in this area.

**Review Assessment: Checking Correctness Of Derivations And Theory:**

I assessed the sensibility of the derivations and theory.

**Review Assessment: Checking Correctness Of Experiments:**

I carefully checked the experiments.

**Review Assessment: Thoroughness In Paper Reading:**

I read the paper at least twice and used my best judgement in assessing the paper.

---

> ### Author Response · Authors · 2019-11-09
> **Response to Reviewer 1**
>
> Thank you for your valuable comments and suggestions.
> _______________________________________________________________________
> “ (1) checking the performance over large datasets (such as ImageNet)”
>
> Thank you for your suggestion. As ImageNet is the largest dataset among the datasets that we have presented in our paper, it may be difficult but we will try our best to run the experiments and report the results before 15th Nov.
> _______________________________________________________________________
> “(2) providing possible analysis on the observation. Otherwise, the readers cannot be fully convinced.”
>
> Section 3 is dedicated to analysing the effect of image transformations on the distribution of softmax, which motivates our method and supports our experimental findings. Following the suggestion from Reviewer 2, we have also added in additional analysis on the divergence of softmax distributions in Fig. 2b, which we believe has improved the analysis in our paper.

---

### Official Review · AnonReviewer2 · 2019-10-29
**Official Blind Review #2**

**Rating:** 3

**Review:**

The authors analyze the use of image transformation as a defense against adversarial examples, where a challenge is to prevent the deterioration of performance on clean images. To do this, they show that the softmax distributions for clean and adversarial images share similar "features", and therefore one can apply a trained distribution classifier which takes the softmax distribution to return the class label. This is as opposed to original approach of making a prediction for each sample (a random transformation of the input image) followed by majority voting.

I have one major concern about their analysis, which makes me lean weakly towards not accepting the paper. I'm happy to change my score after discussions and further clarifications. In particular, I'm not convinced that the phenomena of softmax distributions appearing similarly for clean and adversarial images is not simply a result of the stochastic transformation converging to a fixed (or possibly neighborhood of fixed) distributions. Does the transformation retain information about the difference between classes? That is, while the divergence of softmax distributions from clean and adversarial images decreases as one applies more transformations, does the divergence of softmax distributions from clean images of two different classes also decrease?

As a non-expert in the field, I found the paper well-written. It motivates the idea well by identifying challenges in a promising technique (random tranfsormations) and provides decent background explanation.

Figure 2 is an interesting example showing the relationship of the softmax distributions for clean and adversarial examples. This raises a few questions which are unclear to me:

1. I'm a bit surprised at how remarkably bad transformation-based defenses seem to be in degrading classifier performance. Is it really the case that the MNIST model only gets 78% accuracy on class label 8 and 36% on class label 6? What about without the transformation?
2. This phenomena of softmax distributions being similar seems to largely depend on the stochastic transformation T and form of attack. This is especially seen in Figure 2b, which suggests that T may be acting like a transition kernel which takes an arbitrary initial state and may have it converge to a fixed stationary distribution. What does the divergence look like for two class distributions of clean images over the number of pixel deflections?

For the defense, I'm also curious what happens if the model used the distribution classifier with random transformations at training time, instead of a separate softmax output layer.

Regarding experiments, I only assessed their sensibility. Unfortunately, I don't know enough about the field to tell how significant the results are.

1. For the choice of number of transformations and training set size for the distribution classifier, how were they chosen? They seem to vary arbitrarily across the datasets.
2. It seems like the improvements appear only in single-step FGSM, which also tends to be the worst performing, where the other methods are iterative. What's the intuition for this?


**Experience Assessment:**

I do not know much about this area.

**Review Assessment: Checking Correctness Of Derivations And Theory:**

I assessed the sensibility of the derivations and theory.

**Review Assessment: Checking Correctness Of Experiments:**

I assessed the sensibility of the experiments.

**Review Assessment: Thoroughness In Paper Reading:**

I read the paper at least twice and used my best judgement in assessing the paper.

---

> ### Author Response · Authors · 2019-11-07
> **Clarification on your statement**
>
> Thank you for your detailed comments and insights. Before we respond to the comments, we would like to first clarify something. We appreciate if you can elaborate on the following sentence so that we can understand it better.
>
> “For the defense, I'm also curious what happens if the model used the distribution classifier with random transformations at training time, instead of a separate softmax output layer.”
>
> Thank you.

---

> > ### Comment · AnonReviewer2 · 2019-11-07
> > **re:Clarification on your statement**
> >
> > Thanks for asking. The questions is, can you train the model end-to-end instead of two-stage? If so, can you use a distribution classifier as replacement for the softmax instead of as a two-headed network, and what might be different in terms of the results?

---

> > > ### Author Response · Authors · 2019-11-09
> > > **Response to Reviewer 2, Part 1**
> > >
> > > Thank you for your valuable insights and discussions. We start with addressing your main concern on the divergence of softmax distributions from clean images of different classes, and then addressing the other comments.
> > > _______________________________________________________________________
> > > “In particular, I'm not convinced that the phenomena of softmax distributions appearing similarly for clean and adversarial images is not simply a result of the stochastic transformation converging to a fixed (or possibly neighborhood of fixed) distributions.
> > > Does the transformation retain information about the difference between classes? That is, … does the divergence of softmax distributions from clean images of two different classes also decrease?
> > > …which suggests that T may be acting like a transition kernel which takes an arbitrary initial state and may have it converge to a fixed stationary distribution. What does the divergence look like for two class distributions of clean images over the number of pixel deflections?”
> > >
> > > Yes you raised a very valid point about the random transformation causing a convergence towards a fixed distribution. We have thought of this too but should have articulated it more in our paper. We appreciate your insight which helps us improve the analysis in our updated paper.
> > >
> > > We agree that for certain transformations (eg. random pixel noise), with increasing transformation magnitude, any arbitrary initial distribution will converge to some fixed stationary distribution. The transformation’s transition kernel should be irreducible, in the sense that there exists a path between any two images. Following this argument, divergence of the distributions of softmax between the clean and adversarial images of the same class will decrease, as will the case of clean images from two different classes. The key point is here: for the classification to perform well, the divergence of distributions of softmax from clean images of different classes has to decrease slower than that of clean and adversarial images of the same class. Theoretical study on this convergence rate is a possible future work.
> > >
> > > Following your suggestion, we conducted experiments to study this phenomenon. In Fig. 2b, we have added analysis on divergence of distributions between images from different classes. The curves for all 10 MNIST classes are shown in Fig. 8 in the Appendix. The divergence of softmax distributions from the clean images of two different classes decreases as well. As discussed in the earlier paragraph, this phenomenon is expected due to effects of the random transformation. However, we observe that the divergence of clean images from different classes does not decrease as rapidly and remains higher than the divergence of images from the same class at d=300 pixel deflections. This means the transformation still retains information about the difference between classes. However, at d=800 pixel deflections, all 4 distance measures converge, suggesting that the transformation magnitude is too large and the differences between the classes are no longer retained.
> > >
> > > The key here is whether the subsequent classifier is able to distinguish the different classes. From our experiments, we show that majority voting results in significant deterioration of accuracy for clean images. By using our distribution classifier, we are able to improve on the classification accuracies, with d=300 pixel deflections. For instance, in Fig. 4, clean images E-H were wrongly classified by voting but correctly classified by the distribution classifier.
> > > _______________________________________________________________________
> > > “1. I'm a bit surprised at how remarkably bad transformation-based defenses seem to be in degrading classifier performance. Is it really the case that the MNIST model only gets 78% accuracy on class label 8 and 36% on class label 6? What about without the transformation?”
> > >
> > > Yes your statement is correct, the MNIST model with transformation and majority voting gets 78% accuracy on class 8 and 36% on class 6. Combining all 10 classes together, the average accuracy on clean images is actually 87.6%. It happens that for class 6, the accuracy is much lower compared to the other classes, which means the degradation due to transformation is more serious. In Fig. 2a, the clean images are all classified correctly by the CNN (misclassified images by CNN are excluded), hence without the transformation, the accuracy is 100%. We have added clarifications on the exclusion of clean images misclassified by CNN in Section 3.

---

> > > > ### Author Response · Authors · 2019-11-09
> > > > **Response to Reviewer 2, Part 2**
> > > >
> > > > “For the defense, I'm also curious what happens if the model used the distribution classifier with random transformations at training time, instead of a separate softmax output layer
> > > > …Thanks for asking. The questions is, can you train the model end-to-end instead of two-stage? If so, can you use a distribution classifier as replacement for the softmax instead of as a two-headed network, and what might be different in terms of the results?”
> > > >
> > > > Thank you for your clarification of your question. Yes, in principle, the combined CNN and distribution classifier model could be trained end-to-end with the random transformations. However, the contribution of our method is to train a relatively small distribution classifier model on the CNN softmax outputs, without having to retrain the much larger CNN network. It is a possible future work to compare our more lightweight method with the more costly end-to-end training, and it could be that end-to-end training gives better classification performance.
> > > > _______________________________________________________________________
> > > > “1. For the choice of number of transformations and training set size for the distribution classifier, how were they chosen? They seem to vary arbitrarily across the datasets.”
> > > >
> > > > The number of transformations and training size were chosen by tuning with the validation set. In Fig. 6, we have studied the performance with respect to the number of transformations and chose the value based on that. As for the number of training samples, we found that increasing the training sizes beyond what we reported in our manuscript did not give further improvements.
> > > > _______________________________________________________________________
> > > > “2. It seems like the improvements appear only in single-step FGSM, which also tends to be the worst performing, where the other methods are iterative. What's the intuition for this?”
> > > >
> > > > Improvements of our distribution classifier method over majority voting are actually also evident in other attacks (IGSM, DeepFool, C&W) and the exact numbers are in Tables 11-13. As we mentioned in our analysis in Section 5.1, for IGSM, DeepFool and C&W, baseline voting already has quite good recovery which is in agreement with observations made by Xie et al. (2017) where they found their defense to be more effective for iterative attacks. Hence, the improvement by our distribution classifier is not as drastic as FGSM.

---

### Official Review · AnonReviewer4 · 2019-11-03
**Official Blind Review #4**

**Rating:** 6

**Review:**



===== Post Rebuttal =====

The authors addressed almost all of my comments. I think the revised paper is of higher quality and clarity as a result. Figure 3 is the main remaining concern. However,  given the listed strengths of the paper, I am happy to change my rating to weak accept .

===== Summary =====

One strategy for defense against adversarial attacks is to perturb the input sample multiple times and then aggregate their corresponding outputs into a single output. One recent variant of this method (Prakash et al. 2018) shows using this type of defense mechanism degrades the accuracy on clean (unperturbed) inputs. This paper tries to mitigate the accuracy drop on clean inputs. It does so by learning an additional distribution classifier which takes as input the distribution of perturbed samples’ outputs. The method is trained on MNIST, CIFAR10, CIFAR100 and using 4 different adversarial attacks. It shows significant improvement in several cases over standard input-perturbation methods.


===== Strengths and Weaknesses =====

+ The paper picks an important problem, proposes a simple technique and achieves significant improvements in certain cases.
+ The experiments are done on 3 datasets and 4 different adversarial attack techniques.
+ The final experiment on end-to-end attack is interesting.

Concerns regarding the main paper’s description
- From Figure 1, it seems that the distribution classifier does a binary classification for the distribution of each class separately. If this is the case, how is the final classification done (using the binomial distribution per class)? The figure suggests that the maximum probability of the binary classifications (bus for yellow class with 0.8 score) is taken. This would be strange since they are not optimized to compete with each other (e.g. as in a more proper cross entropy). In any case, the details of the distribution classifier is missing from the paper, and section 4 refers to the figure. The details should come in the main text using precise language and math formulation.

- page 4 is very hard to follow since it tries to give a textual description to mathematical concepts. I think it is important to provide the mathematical definition such that 1) it is easier to follow 2) the concepts are unambiguously defined. This includes a definition of softmax distribution, joint distribution of output and transformation, marginalized distributions, clean intra-class distance, adversarial intra-class distance, and clean-adversarial inter-class distance.

- page 4: it seems it is implied in the definition of “clean” images that these images are classified “correctly” by the network. Otherwise, in Figure 2.a, the clean row is uninterpretable not knowing whether the clean image was classified correctly or not.

- page 4,5: what is a “transformation magnitude”? It could refer to both the number of transformations and also the number of perturbed pixels. The behavior in Figure 3 suggests that the latter should be the case, but it’s important to disambiguate.

- figure 3 is counterintuitive. The support (possible distributions) should normally live on a simplex (due the sum-to-one probability constraint). In figure 3, it seems there is support in the full unit hypercube. In particular, we can see density along the horizontal/vertical line of 1-density for one class (i.e. non-zero density for others in this case)

- page 6: “[...] to train a distribution classifier on the distributions obtained from clean images only [...]“. I cannot follow the argument at the end of the first paragraph that led to the proposed *training on clean images only*. I understand that from the qualitative demonstration of 8 different examples in Figure 4 the distributions of perturbed adversarial examples *might be* redundant to the perturbed wrongly-classified clean examples. However, even if it is redundant, why should it be harmful to train the distribution classifier with adversarial examples as well as clean examples?

Regarding Experiments:
- why is the evaluation done only based on accuracy of *correctly-classified* clean images and the recovery of *wrongly-classified* adversarial images? Couldn’t it be that the baseline (majority voting) does a better job than the distribution classifier on the “wrongly-classified” clean images and the “correctly-classified” adversarial images?
- I think it's still interesting and informative to train the distribution classifier using both clean and perturbed inputs to compare the performance.
- page 6: why is there a need for applying a kernel density estimator followed by a discretization (e.g. for instance instead of a simple histogram)?

===== Final Decision =====
My current “weak reject” rating is primarily based on the unclarity of the main paper and secondary regarding the concerns for experiments.

===== Points of Improvement =====
I believe the paper will become stronger if the proposal is described and motivated more formally.



**Experience Assessment:**

I do not know much about this area.

**Review Assessment: Checking Correctness Of Derivations And Theory:**

I assessed the sensibility of the derivations and theory.

**Review Assessment: Checking Correctness Of Experiments:**

I assessed the sensibility of the experiments.

**Review Assessment: Thoroughness In Paper Reading:**

I read the paper at least twice and used my best judgement in assessing the paper.

---

> ### Author Response · Authors · 2019-11-09
> **Response to Reviewer 4, part 1**
>
> Thank you for your detailed comments. We have improved our presentation of the paper based on your suggestions.
> _______________________________________________________________________
> “From Figure 1, it seems that the distribution classifier does a binary classification for the distribution of each class separately… This would be strange since they are not optimized to compete with each other (e.g. as in a more proper cross entropy). In any case, the details of the distribution classifier is missing from the paper…”
>
> We have realized that Figure 1 can be misleading and we apologize for the confusion. We have changed it in our updated paper. It is not true that the distribution classifier does a binary classification for each class separately, and we apologize for leading you to think this way. In our previous submitted version, we had wanted to show the distribution classifier specific for the case of DRN only, which led to some confusion. We have edited Fig. 1 to reflect the case of a general distribution classifier, which is valid for DRN, RF and MLP. To train the classifiers, we use the appropriate cross entropy loss for DRN and MLP, and the Gini impurity is used as the splitting criterion in RF. We have added this detail in our main text in Section 4.
> _______________________________________________________________________
> “page 4 is very hard to follow since it tries to give a textual description to mathematical concepts. I think it is important to provide the mathematical definition...”
>
> Yes we agree that mathematical definitions are important. The following definitions were already included in the Appendix in our previous version and we have shifted them to the main text in the updated version:
> 1.	The mathematical description of obtaining the distribution of softmax, which is the marginal distributions over each class’ softmax output, is shifted from the Appendix to start of Section 3 (see Eqn. (1))
> 2.	Distance metrics for Fig. 2b is shifted from the Appendix to page 5, with added mathematical definitions
> _______________________________________________________________________
> “page 4: it seems it is implied in the definition of “clean” images that these images are classified “correctly” by the network. Otherwise, in Figure 2.a, the clean row is uninterpretable not knowing whether the clean image was classified correctly or not.”
>
> Yes, “clean” images are all the images that are classified correctly by the CNN.  We added clarifications on the definition of “clean” images at the start of Section 3. We did not include the misclassified images by CNN because 1) following Prakash et al. (2018), it is not meaningful to attack images that are already misclassified and 2) the CNN has high test accuracy on the clean images so misclassifications are the minority (98.7% test accuracy, with reference to Fig. 2a).
> _______________________________________________________________________
> “page 4,5: what is a “transformation magnitude”? It could refer to both the number of transformations and also the number of perturbed pixels. The behavior in Figure 3 suggests that the latter should be the case, but it’s important to disambiguate.”
>
> You are right that in Section 3, transformation magnitude refers to the number of pixel deflections. The number of transformed samples per image for MNIST is kept at N=100 throughout the paper. We have clarified this at the start of Section 3 and replaced ‘transformation magnitude’ with ‘number of pixel deflections’ to disambiguate the terms.
> _______________________________________________________________________
> “figure 3 is counterintuitive. The support (possible distributions) should normally live on a simplex (due the sum-to-one probability constraint). In figure 3, it seems there is support in the full unit hypercube. In particular, we can see density along the horizontal/vertical line of 1-density for one class (i.e. non-zero density for others in this case)”
>
> Yes you are correct that the support of the distributions should live on a simplex, which in this MNIST case is simplex in a 10-dimensional space. However, as we have mentioned in our main text, for visualization purposes, we show the softmax values for 2 chosen dimensions (class 5 and class 6), which is equivalent to projecting from 10 dimensions to 2 dimensions.  Hence, the density along the horizontal/vertical line means that at least one of the other 8 classes have non-zero softmax values, and the sum of softmax values for class 5 and 6 can be less than 1. We have enhanced the explanations for interpreting Fig. 3 in the caption and in the main text.

---

> > ### Author Response · Authors · 2019-11-09
> > **Response to Reviewer 4, Part 2**
> >
> > “page 6: …I cannot follow the argument at the end of the first paragraph that led to the proposed *training on clean images only*…
> > …However, even if it is redundant, why should it be harmful to train the distribution classifier with adversarial examples as well as clean examples?...”
> >
> > There is an important reason why training just on clean images is preferable. In real-world settings, the type of attack is not known beforehand. Training the distribution classifier on a specific attack may cause the classifier to overfit to that attack. Hence, it is an advantage that our defense method is attack-agnostic. To emphasize the above reasons, we have added in explanations in our introduction section.
> >
> > We have previously tried training the distribution classifier (DC) with clean and adversarial images which we term as lightweight adversarial training. We found that it indeed overfits to the attack and does not generalize across all attacks. Here we show the results for MNIST, with DRN trained on different adversarial attacks, and test on IGSM adversarial images with random resize and padding (RRP) transformation.
> >
> > Transformation (RRP) + DC (DRN) 	Test accuracy on IGSM images (%)
> >    	   Clean only	       			                   82.4 (Table 11, last column under IGSM, DRN)
> >  	 Clean+IGSM         			           88.1
> >   	Clean+FGSM				                   83.0
> >       Clean+DeepFool			                   84.4
> >  	 Clean+C&W				                   78.9
> >
> > Compared to training on clean images only, training on a mixture of clean and adversarial IGSM images improved the accuracy on adversarial IGSM images (82.4% to 88.1%). However, when the distribution classifier is trained on other attacks, the classifier does not always generalize well to IGSM (eg. overfits on C&W adversarial images with 78.9% accuracy).
> > _______________________________________________________________________
> > “- why is the evaluation done only based on accuracy of *correctly-classified* clean images and the recovery of *wrongly-classified* adversarial images? Couldn’t it be that the baseline (majority voting) does a better job than the distribution classifier on the “wrongly-classified” clean images and the “correctly-classified” adversarial images?”
> >
> > First, following Prakash et al., 2018, we did not perform attack or defense on images initially misclassified by the CNN as it does not meaningfully show the effectiveness of the attack and subsequent defense. For the same reason, we did not perform evaluation on the images that the attack has failed (which you refer to as “correctly-classified” adversarial images). We would like to highlight that by definition, adversarial images are images that have been attacked successfully to be misclassified by the CNN.
> >
> > Second, the CNN is expected to perform well on the clean images, so the number of “wrongly-classified” clean images is actually small (eg. CNN’s MNIST test accuracy is 98.7%). Also, most of the attacks have very high success rate of above 90% so the “correctly-classified” adversarial images are the minority.
> >
> > As an additional check, we performed evaluations on clean images misclassified by CNN and images that the attack has failed, and find that the distribution classifier still performs better than baseline majority voting. The results below are for CIFAR100, with FGSM attack, with random resize and padding transformation and trained with random forest classifier.
> > 					         		                                 Test accuracy (%)
> > 					                          No transformation | Voting | Distribution classifier
> > Clean images misclassified by CNN		         0	             |   12.3   |    18.1
> > Images where attack on CNN failed		100	             |    65.7  |    79.5
> > _______________________________________________________________________
> > “- page 6: why is there a need for applying a kernel density estimator followed by a discretization (e.g. for instance instead of a simple histogram)?”
> >
> > We use kernel density estimate as it provides a principled way to account for the uncertainty in the random image transformation, where the kernel width (tuned by cross validation) correlates to the extent of uncertainty. Furthermore, a practical reason is that using histogram binning may result in empty bins especially when number of samples is small. It also causes discontinuities in the estimation of the probability distribution.

---

> > > ### Comment · AnonReviewer4 · 2019-11-15
> > > **Outstanding points after the rebuttal**
> > >
> > > I appreciate the authors’ efforts in addressing the concerns. It alleviates most of my concerns including the clarity of the presentation. The two outstanding clarifications:
> > >
> > > - The rebuttal for Figure 3, does not explain my main concern that is how can a point which corresponds to a sum probability of larger than one have non-zero density.
> > >
> > > - It might be due to my ignorance of the field, but I still don’t see that “correctly-classified” images and unsuccessful attacks are irrelevant. The argument that it is not meaningful to attack images that are already misclassified is understandable in most cases (although one might still want to change the maximum class). However, defense (of the proposed form) is supposedly applied to *all examples*, since, as far as I understand, it cannot be a priori known if the input is a successful attacked input or a clean image. In that regard, it’s important that the defense system does not reduce the performance on the clean images, correctly classified or not. Same argument goes for unsuccessful adversarial attacks -- they cannot be known apriori and thus important to keep/improve the performance on those when defending.
> > >
> > > Regarding the second point, the authors provide results showing that it works better even for these two cases which is encouraging! But, I think it's important to include this discussion and numbers in the paper.

---

> > > > ### Author Response · Authors · 2019-11-15
> > > > **Response to the outstanding points**
> > > >
> > > > Thank you for your additional feedback, here are our responses to the outstanding points.
> > > >
> > > > "- The rebuttal for Figure 3, does not explain my main concern that is how can a point which corresponds to a sum probability of larger than one have non-zero density."
> > > >
> > > > Thank you for clarifying this point. The densities shown in Figure 3 are for visualization purposes only, these do not affect our main experimental results. The purpose of this visualization is to explain the changes in the distribution shapes of clean and adversarial images with the image transformation. We perform kernel density estimation (kde) on the softmax values for the marginals on class 5 and 6. We have not excluded the areas where performing kde results in sum probability exceeding one, and our visualization still conveys our ideas and the distribution shapes well. Here we would like to note that this density estimation done for visualization in Figure 3 is different from the density estimation for our distribution classifier described in Section 3. We have added more explanations on this point in our paper (in 4th paragraph of page 5).
> > > >
> > > > _______________________________________________________________________
> > > >
> > > > "...However, defense (of the proposed form) is supposedly applied to *all examples*, since, as far as I understand, it cannot be a priori known if the input is a successful attacked input or a clean image. ... Regarding the second point, the authors provide results showing that it works better even for these two cases which is encouraging! But, I think it's important to include this discussion and numbers in the paper."
> > > >
> > > > Yes we agree that the defense system does not know a priori whether the input is a clean image (whether correctly classified or not) or an attacked image (whether successful or not), and should perform well for all cases. Following your suggestion, we have included the discussion in the main text (Section 5.2 paragraph 2, page 9). Since we have presented the results only for CIFAR100 (with FGSM attack, random resize and padding and random forest), we have included the specific numbers in the Appendix in Table 14.

---

### Author Response · Authors · 2019-11-09
**To all reviewers**

We thank all reviewers for their detailed reviews and constructive comments. Our paper has been updated following your suggestions. Please refer to the individual reviews for our responses.

Here are the main updates to our paper:
1. Mathematical definitions of distribution of softmax (see Eqn. (1)) have been shifted from Appendix to start of Section 3
2. Mathematical definitions for distance metrics for Fig. 2b have been shifted from Appendix to page 5
3. Following Reviewer 2's suggestion, in Fig. 2b, we have added analysis on divergence of distributions between images from different classes

Thank you for your time.

---

> ### Author Response · Authors · 2019-11-13
> **New updated version of paper**
>
> We have made minor corrections for spelling in our latest version.

---

### Decision · Program_Chairs · 2019-12-19

**Decision:**

Accept (Poster)

**Comment:**

This paper investigates tradeoffs between preserving accuracy on clean samples and increasing robustness on adversarial samples by using transformations and majority votes. Observations on the distribution of the induced softmax show that existing methods could be improved by leveraging information from that distribution to correct predictions, as confirmed by experiments.
The problem space is important and reviewers find the approach interesting. Authors have provided some necessary clarifications during rebuttal and additional experiments. While some reservations remain, this paper's premise and its experimental results appear sufficiently interesting to justify an acceptance recommendation.